# Nutrition Assessment and Adverse Outcomes in Hospitalized Patients with Tuberculosis

**DOI:** 10.3390/jcm10122702

**Published:** 2021-06-18

**Authors:** Huang-Shen Lin, Ming-Shyan Lin, Ching-Chi Chi, Jung-Jr Ye, Ching-Chuan Hsieh

**Affiliations:** 1Division of Infectious Diseases, Department of Internal Medicine, Chang Gung Memorial Hospital, Chiayi 61363, Taiwan; 2College of Medicine, Chang Gung University, Taoyuan 33303, Taiwan; chingchi@cgmh.org.tw (C.-C.C.); Jeffrey570404@gmail.com (C.-C.H.); 3Division of Cardiology, Chang Gung Memorial Hospital, Chiayi 61363, Taiwan; mingshyan@gmail.com; 4Department of Dermatology, Chang Gung Memorial Hospital, Linkou, Taoyuan 33305, Taiwan; 5Division of Infectious Diseases, Department of Internal Medicine, Chang Gung Memorial Hospital, Keelung 204, Taiwan; loyalwise@gmail.com; 6Department of Nutrition, Chang Gung Memorial Hospital, Chiayi 61363, Taiwan

**Keywords:** adverse outcomes, hepatitis, liver injury, malnutrition, nutritional assessment, Tuberculosis

## Abstract

Background: Malnutrition in patients with tuberculosis (TB) is associated with poor outcomes. This study assessed the validity of the patient-generated subjective global assessment (PG-SGA) in adult TB patients and examined the association of the PG-SGA score with adverse outcomes. Methods: This is a retrospective chart review study compared with the well-nourished and malnourished TB patients. The nutritional status was determined using the PG-SGA for adult patients (*n* = 128). Clinical outcomes included liver injury and mortality. Adverse outcomes included hepatitis during anti-tuberculosis therapy. Results: By comparing nutritional status using global assessment, well-nourished patients had a significantly higher body weight index (*p* = 0.002), a lower PG-SGA score (*p* < 0.001), and lower diabetic rate (*p* = 0.029). Malnourishment was a risk factor (*p* = 0.022) for liver injury and fatal outcomes (*p* < 0.001). A higher PG-SGA score was a risk factor for liver injury (*p* = 0.002) and an independent risk factor for fatal outcomes (*p* = 0.031). ROC analysis for outcome prediction showed that a PG-SGA score of 5.5 points yielded the most appropriate sensitivity (61.5%) and specificity (64.7%). Conclusion: Both global assessment and the total PG-SGA score were related to tuberculosis outcome and liver injury during anti-TB treatment.

## 1. Introduction

Tuberculosis (TB) remains a major worldwide problem. In 2019, an estimated 10.0 million new cases occurred globally and there were 1.4 million TB deaths [1]. The relationship between TB and malnutrition has long been discussed because it is believed that malnutrition results in a predisposition for developing clinical diseases and that TB contributes to malnutrition [2,3,4]. In patients undergoing anti-TB treatment, malnutrition may not only advance the disease but also cause a higher mortality rate [5,6,7]. Presently, nutritional support is considered an important component of TB therapy [8].

Nutritional evaluation was divided into two types: objective and subjective. The objective type included hematological, biochemical, and anthropometric evaluation such as body mass index (BMI); the subjective type included nutrition impact symptoms and physical functioning such as patient-generated subjective global assessment (PG-SGA).

PG-SGA consists of a total score and global assessment [9]. The total PG-SCA score is a continuous measure and is the sum of the scores of seven questions about weight loss, food intake, symptoms, activities, disease, metabolic demand, and a physical examination. Global assessment categories include weight, nutrient intake, nutrition impact symptoms, physical functioning, and a physical examination. The patients would be divided into 3 groups: well-nourished (SGA-A), moderately nourished or suspected of being malnourished (SGA-B), and severely malnourished (SGA-C).

PG-SGA is specifically used to evaluate patients with cancer [10,11] and patients with other chronic diseases [12,13]. We evaluated the validity of the PG-SGA as a nutritional assessment tool in adult patients with TB, a chronic infectious disease, and the association of PG-SGA with adverse effects and outcomes.

## 2. Methods

### 2.1. Study Design and Patients

Chang Gung Memorial Hospital in Chia-Yi, Taiwan, is a 1200 bed tertiary teaching hospital. A retrospective study was conducted between 1 June 2010 and 31 December 2013 on consenting adult inpatients with positive *Mycobacterium tuberculosis* complex (MTB) cultures from specimens obtained during admission. All patients were treated with an initial therapeutic regimen of rifampin, isoniazid, and pyrazinamide (RIP) with or without ethambutol (if RIP all sensitive) therapy. Patients who were HIV-positive, in critical condition, or initially admitted to the intensive care unit were excluded. The data were analyzed anonymously. This study was approved by the Institutional Review Board of the Chang Gung Memorial Hospital (Number: 104-2837B).

### 2.2. Demography, Comorbidity, and Constitutional Symptoms

Data on age, gender, and comorbid conditions were obtained from the patients’ medical records. Patients ≥65 years old were classified as “elderly” and patients between 18 and 64 years old were considered “non-elderly”. Verified comorbidities included diabetes mellitus (DM), heart disease, chronic obstructive pulmonary disease (COPD), chronic kidney disease (CKD) stage V, decompensated liver cirrhosis (DLC), solid organ or hematological malignancy, and hematology disease. Constitutional symptoms included fever (body temperature ≥38 °C) and body weight loss (weight loss ≥5% of the initial weight in 6 months).

### 2.3. Diagnosis

Patients were diagnosed with TB if they had a positive culture with mycobacterial tuberculosis (MTB) and patients showing pathogen resistance to RIF and high-level resistance to INA (10 mg/L) were excluded from the study.

Pulmonary TB was considered if lower respiratory tract cultures were positive for MTB, including cultures from sputum, tracheal aspirates, bronchoalveolar lavage fluid, and lung tissue. Extrapulmonary TB included pleural TB (pleural effusion or pleural tissue cultures positive for MTB), TB lymphadenitis (lymph node cultures positive for MTB), TB arthritis (synovial fluid cultures positive for MTB), spinal TB (paraspinal abscess, epidural abscess, or spine bone tissue cultures positive for MTB), and other types (determined by positive MTB cultures from corresponding organ systems). Disseminated TB was defined as TB involvement in at least two organ systems.

### 2.4. Treatment, Adverse Drug Reactions, and Outcomes

Liver injury is the most common and important potential adverse reaction after using RIP. Liver injury is diagnosed if one of the following conditions is met: (1) transaminase level >5 times the normal upper limit; (2) transaminase level >3 times the normal upper limit with liver injury associated symptoms; (3) total bilirubin level >3 mg/dl with liver injury associated symptoms. Patients who underwent a full course of TB treatment with good clinical responsiveness and microbiological eradication were considered to have had favorable outcomes. Patients who died during the therapeutic course were classified as having fatal outcome.

### 2.5. Patient-Generated Subjective Global Assessment

Experienced dietitians assessed patients’ nutritional status within 24 h of their admission, using both the PG-SGA and anthropometric and biochemical parameters. The PG-SGA consisted of a medical history completed by patients using a check-box format and physical examinations were conducted to evaluate fat, muscle stores, and fluids. Responses provided by the patients were initially corroborated by the examiner, who subsequently provided a global rating of well nourished (SGA-A), moderately nourished or suspected of being malnourished (SGA-B), or severely malnourished (SGA-C). Numerical scores (from 0 to 35) were allocated for each tool component, including weight, food intake, symptoms, activities, disease, metabolic demand, and physical examination and the scores were subsequently summed [14].

### 2.6. Statistical Analysis

SPSS 18.0 for Windows (SPSS Inc., Chicago, IL, USA) was used for all statistical analyses. Categorical variables were analyzed using the χ^2^ test or Fisher’s Exact test, as appropriate, and continuous variables were analyzed using independent *t* tests. Odd ratios (ORs) and 95% confidence intervals (CIs) were calculated. Variables with *p*-value < 0.1 in univariate analysis and variables of interest were included in a logistic regression model for multivariate analysis. All tests were two-tailed and significance was set at *p* < 0.05 in multivariate analysis. Receiver operator characteristic (ROC) analysis was used to examine the validity of the PG-SGA score to predict patient outcomes.

## 3. Results

### 3.1. Demography, Comorbidities, and Clinical Conditions

We enrolled 128 patients with TB in our study (mean age: 70.7 years; male: 94 (73.4%)) (Table 1). The mean BMI was 21.4 kg/m^2^ and the mean PS-SGA score was 5.2. Diabetes mellitus (DM) was the most common comorbid disease (23.4%) followed by solid tumor (18.0%). Initially, 34 patients (26.6%) had fever and 10 patients (7.8%) had body weight loss (Table 1). For most patients, the source of the infection was pulmonary (91.4%). There were 19 patients with extrapulmonary TB (four with pleural TB, three with spinal TB, three with TB lymphadenitis, three with TB arthritis, two with urinary tract TB, one with TB pericarditis, one with TB peritonitis, one with TB meningitis, and one with urinary tract TB combined with spinal TB). Fifty-five patients (43%) had positive results for acid-fast staining.

### 3.2. Well-Nourished and Malnourished Patients Compared

When patients were grouped according to their SGA ratings, BMI was significantly lower in malnourished than in well-nourished patients (*p* = 0.002) and PG-SGA scores were also significantly lower in malnourished patients (*p* < 0.001). The well-nourished group had a lower ratio of patients with DM than did the malnourished group (15.4% vs. 31.7%; *p* = 0.029) (Table 1).

Twenty-six patients (20.3%) had liver injury during the course of treatment. The malnourished group had a higher ratio of liver injury during therapy (28.6% vs. 12.3%; *p* = 0.022) and fatal outcomes (33.3% vs. 7.7%; *p* < 0.001) compared with the well-nourished group. Comparing the liver-injury-negative group, the liver-injury-positive group had a higher ratio of patients with heart disease (only two participants) (7.7% vs. 0.0%; *p* = 0.004) and had higher PG-SGA scores (7.3 ± 3.6 vs. 4.7 ± 3.5; *p* = 0.002) (Table 2). The ROC analysis for outcome prediction showed that a PG-SGA score of 5.5 points yielded the most appropriate sensitivity (61.5%) and specificity (64.7%). Moreover, patients with PG-SGA scores ≥ 6 had a significantly higher incidence of liver injury (32.7% vs. 11.8%; *p* = 0.007) and fatal outcomes (30.8% vs. 13.2%; *p* = 0.024) than did patients with lower PG-SGA scores (Figure 1).

### 3.3. Factors Associated with Fatal Outcomes

There were 26 patients (20.3%) with fatal outcomes in the study group. In univariate analysis, patients with fatal outcomes had higher rates of DM (*p* = 0.043) and higher PG-SGA scores (*p* = 0.028). However, fatal outcomes were not significantly associated with BMI (favorable vs. fatal outcome: 21.6 ± 4.1 vs. 20.9 ± 4.4, *p* = 0.467) or BMI < 18.5 (9 (34.6%) vs. 26 (26.0%), *p* = 0.525). In multivariate analysis, the PG-SGA score was the only independent predictor of fatal outcomes (*p* = 0.031) (Table 3). The area under the ROC curve for PG-SGA scores compared with outcome (favorable vs. fatal outcome) was 0.649 (Figure 2). In this analytical model, a cut-off value of 5.5 points yielded the most appropriate sensitivity (61.5%) and specificity (64.7%).

## 4. Discussion

Solid epidemiological evidence indicates that poor nutritional status is a risk factor for tuberculosis and malnutrition might also be a risk factor for poor outcomes in those patients with TB. Our study demonstrated a significant correlation between PG-SGA scores and outcome for anti-TB management. Patients with TB and higher PG-SGA scores had a risk 1.142 times larger (95% CI: 1.012–1.288; *p* = 0.031) for fatal outcomes than do patients with TB and lower PG-SGA scores in our analysis.

### 4.1. Nutrition, Diabetes, and Tuberculosis

The World Health Organization guidelines suggest that everyone with active TB should have their nutritional status assessed and then be appropriately counseled, based on that nutritional status, at diagnosis and throughout anti-TB treatment [8]. Malnutrition such as smoking, alcohol abuse, diabetes, and Human Immunodeficiency Virus infection has contributions to the attributable risk for tuberculosis. The consequences of malnutrition may include decreased response to treatment, a delay in recovery, a compromised immune system, increased susceptibility to infection, a lower quality of life, and an increased risk of death in many patients [15]. In this study, patients with malnutrition had typical features: a significantly lower body weight, BMI, and PG-SGA scores and a higher incidence of DM. Globally, 15% of tuberculosis cases are estimated to be attributable to DM [16]. In our data, 23.4% of all patients had DM and 31.7% had malnutrition. We found, in univariate analysis, that DM is a risk factor for poor outcomes. TB induces active infection, malnutrition, and insulin resistance and diabetes is further associated with impaired immunity, increased risk of tuberculosis treatment failure, death, and late relapse [17].

### 4.2. Limitations of BMI Assessment for Elderly Patients

Many tools for nutritional assessment, such as the BMI and PG-SGA score, have been used to evaluate adverse drug reactions and clinical outcomes [9,18]. BMI was the simplest and most common and it was considered related to the outcome of patients with TB [6,19]. In this study, however, unlike the PG-SGA score, it was not significantly related to fatal outcomes or liver damage. A large percentage of patients (71.9%) were elderly. Advanced pathological change such as vertebral flattening, fractures, compression, and attrition of intervertebral discs, dorsal kyphosis, scoliosis, bowing of legs, flattening of the plantar arch, and being bedridden could result in inaccurate height measurements and the overestimations of BMI [20,21]. Bhargava et al. [6] showed a higher incidence of moderate-to-severe malnutrition (BMI < 17.0 kg/m^2^) in the ill population (80% women and 67% men) and Zachariah [18] reported that the incidence of patients with malnutrition at admission (BMI < 18.5 kg/m^2^) was 57%. However, only 28.1% of our patients had a BMI < 18.5 kg/m^2^ and the percentage was significantly lower than in other studies. Therefore, the BMI was not significant for predicting fatal outcomes in the study, despite the claim by Hassen Ali et al. [22] that a low BMI (BMI < 18.5 kg/m^2^) was an independent risk factor for hepatotoxicity during the intensive phase of TB treatment in his database and as claimed by other two other studies [23,24].

### 4.3. PG-SGA Scores, Hepatitis, and TB

In patients with chronic diseases, PG-SGA scores provide a guideline for the level of nutrition intervention required, as well as facilitates quantitative outcome data collection [14]. The PG-SGA is based on a combination of known prognostic indicators, such as weight loss and performance status, and the clinical aspects of dietary intake and nutrition impact symptoms, which were strongly associated with chronic TB infection. PG-SGA is a simple, bed-side, and acute evaluation of nutrition status, which is essential for patients with TB who are beginning poly-pill anti-TB therapy. Comparing global assessment categories, PG-SGA is a continuous measure with multiple-level analysis for nutrition status. Many patients had gastrointestinal irritation complicated with poor appetite, nausea, vomiting, and anorexia during their first weeks of therapy. Isoniazid is a vitamin B6 (pyridoxine) antagonist and rifampin might interfere with folate as well as vitamin B12. Both will disturb digestion and metabolism during therapy. In addition, malnutrition is one of major contributing factors associated with hepatitis during anti-TB treatment [25,26] and our study showed that patients with higher PG-SGA scores had higher incidences of hepatitis and poor outcomes.

Nutritional supplementation in TB prevention and health outcomes has been discussed for the past 20 years and the World Health Organization guidelines^8^ for nutritional care and support for patients with tuberculosis were made public in 2013. The report suggested that all individuals with active TB should (i) be assessed for nutritional status and (ii) be given appropriate counseling based on nutritional status at diagnosis and throughout treatment. Our study supported PG-SGA as a method for evaluating nutritional status that is also related to outcome and adverse effects. In patients with cancer, PG-SGA scores provide a guideline for the level of nutrition intervention required and facilitate the collection of quantitative outcome data [14]. A score of > 9 indicates a critical need for nutrition intervention and scores in the 4–8 range indicate that intervention by a dietitian in conjunction with nurse or physician is required. In this study, for patients with TB, the core with PG-SGA scores > 6 tended to have highly fatal outcomes and liver damage. Additional research and discussion are required to determine optimal levels and methods of nutrition intervention.

### 4.4. Limitations

This study had some limitations. First, this study enrolled only inpatients and the participants diagnosis in the outpatient department were not be included. Moreover, all patients had their nutrition status evaluated within 24 h of being admitted for TB and were placed in standard isolation for anti-TB medication. Second, the total etiologies of liver injury during anti-TB therapy were not well defined in this study and liver injury in the study may not have been restricted to drug-induced liver injury. Third, the sample size in our study was not large; thus, large prospective studies with long-term follow-ups are recommended. Previous studies with small case numbers considered the SGA, which is a nutrition assessment tool and may be a useful for pulmonary TB patients [7]. In this study, we firstly evaluated the relationship between PG-SGA score and TB in a single center and we included all hospitalized patients during the follow-up. We hope further studies may assess and evaluate the PG-SGA score not only in the initiation tuberculosis therapy but also during therapeutic course. By utilizing seriousness assessment, we may establish the relationship of therapeutic outcomes including clinical cure, microbiology eradication, adverse events, and the dynamic PG-SGA scores of the TB patients.

## 5. Conclusions

In conclusion, our findings suggest that there is an alternative to BMI for evaluating nutrition status and for predicting the outcomes and clinical characteristics of patients with TB. Total PG-SGA score and global assessment (two different parts of the PG-SGA) are significantly related to the outcomes of TB and liver injury during anti-TB treatment.

## Figures and Tables

**Figure 1 jcm-10-02702-f001:**
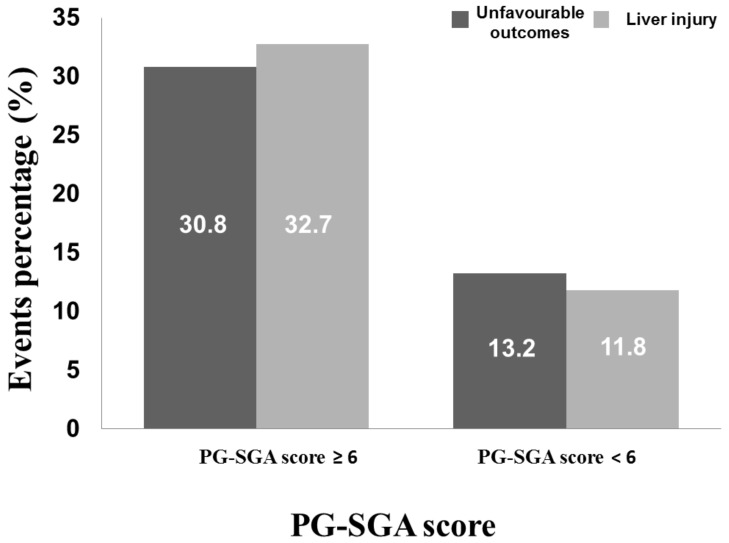
The incidence of liver injury and fatal outcomes according to patient-generated subjective global assessment (PG-SGA) scores.

**Figure 2 jcm-10-02702-f002:**
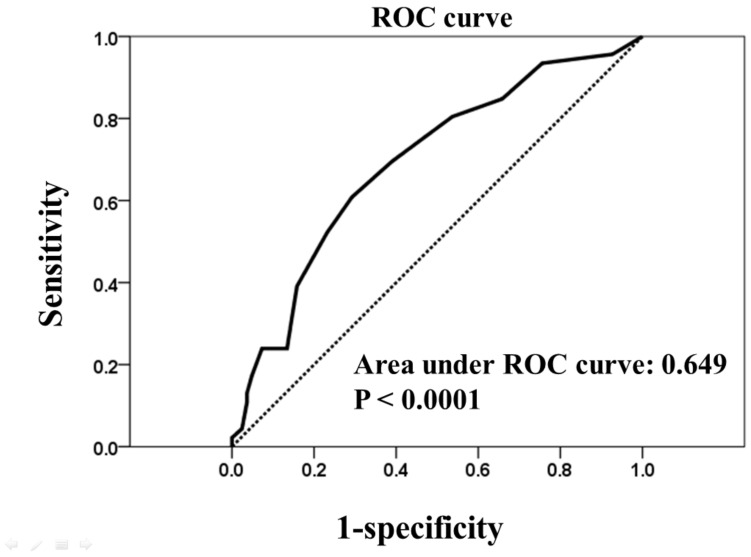
Receiver operator characteristic (ROC) curve for patient-generated subjective global assessment (PG-SGA) compared with outcomes (favorable outcome vs. fatal outcome). The 45° line represents the curve for an ROC area of 0.5. The area under the curve (AUC) is 0.649; 95% confidence interval: 0.530–0.767, thus indicating the PG-SGA score.

**Table 1 jcm-10-02702-t001:** Clinical characteristics and comparison analysis of well-nourished and malnourished patients with tuberculosis treated with regimens containing rifampicin, isoniazid, and pyrazinamide.

	All ^a^	Well-Nourished ^a^	Malnourished ^a^	
Characteristics	*n* = 128	*n* = 65	*n* = 63	*p*
Female ^a^, *n* (%)	34 (26.6)	18 (27.7)	16 (25.4)	0769
Elderly ^b^, *n* (%)	92 (71.9)	46 (70.8)	46 (73.0)	0.777
Weight (kg) ^c^	55.3 ± 11.7	58.4 ± 11.2	52.1 ± 11.4	0.002
Body mass index (kg/m^2^) ^c,d^	21.4 ± 4.2	22.5 ± 3.8	20.3 ± 4.3	0.002
Body mass index (kg/m^2^) <18.5 ^d^	35 (27.7)	11 (16.9)	24 (38.4)	0.007
PG-SGA score ^c^	5.2 ± 3.7	2.6 ± 1.8	7.8 ± 3.2	<0.001
Hemoglobin (mg/dL) ^c^	11.6 ± 2.6	11.9 ± 2.7	11.4 ± 2.4	0.325
Comorbidity, *n* (%)				
Diabetes mellitus	30 (23.4)	10 (15.4)	20 (31.7)	0.029
Heart disease	2 (1.6)	0 (0.0)	2 (3.2)	0.240
COPD	13 (10.2)	8 (12.3)	5 (7.9)	0.413
CKD stage 5	6 (4.7)	3 (4.6)	3 (4.8)	1.000
DLC	3 (2.3)	2 (3.1)	1 (1.6)	1.000
Solid tumor cancer	23 (18.0)	10 (15.4)	13 (20.6)	0.439
Hematological malignancies	5 (3.9)	1 (1.5)	4 (6.3)	0.204
Constitutional symptom, *n* (%)				
Fever	34 (26.6)	14 (21.5)	20 (31.7)	0.191
Body weight loss	10 (7.8)	5 (9.2)	5 (9.5)	1.000
Infectious source, *n* (%)				
Pulmonary	117 (91.4)	58 (89.2)	59 (93.7)	0.372
Extrapulmonary	19 (14.8)	11 (16.9)	8 (12.7)	0.502
Dissemination	8 (6.3)	4 (6.2)	4 (6.3)	1.000
Smear positive	55 (43.0)	25 (38.5)	30 (47.6)	0.295
Liver injury during therapy, *n* (%)	26 (20.3)	8 (12.3)	18 (28.6)	0.022
Fatal outcome, *n* (%)	26 (20.3)	5 (7.7)	21 (33.3)	<0.001

^a^ Data are presented as *n* (%). ^b^ Elderly ≥ 65 years old. ^c^ Data are presented as mean ± SD. ^d^ BMI data were available for 126 patients, including 65 in the well-nourished group and 61 in the malnourished group. CKD: chronic kidney disease; COPD: chronic obstructive pulmonary disease; DLC: decompensated liver cirrhosis; TB: tuberculosis.

**Table 2 jcm-10-02702-t002:** Risk factors for liver injury during treatment.

	Liver Injury ^a^	Non-Liver Injury ^a^	
Variables	*n* = 26	*n* = 102	*p*
Female, *n* (%)	5 (19.2)	29 (28.4)	0.343
Elderly ^b^, *n* (%)	19 (73.1)	73 (71.6)	0.879
Body mass index (kg/m^2^) ^c,d^	20.7 ± 3.0	21.6 ± 4.4	0.204
Body mass index (kg/m^2^) < 18.5 ^d^	8 (30.8)	27 (27.0)	0.717
PG-SGA score ^c^	7.3 ± 3.6	4.7 ± 3.5	0.002
Hgb ^c^	12.3 ± 2.6	11.5 ± 2.6	0.166
Comorbidity, *n* (%)			
Diabetes mellitus	5 (19.2)	25 (24.5)	0.571
Heart disease	2 (7.7)	0 (0.0)	0.004
COPD	4 (15.4)	9 (8.8)	0.299
CKD stage 5	1 (3.8)	5 (4.9)	1.000
DLC	1 (3.8)	2 (2.0)	0.497
Solid tumor cancer	2 (7.7)	21 (20.6)	0.160
Hematological malignancies	0 (0)	5 (4.9)	0.582
Constitutional symptom, *n* (%)			
Fever	7 (26.9)	27 (26.7)	0.963
Body weight loss	1 (3.8)	9 (8.8)	0.686
Infectious source, *n* (%)			
Pulmonary	24 (92.3)	93 (91.2)	1.000
Extrapulmonary	3 (11.5)	16 (15.7)	0.762
Dissemination	1 (3.8)	7 (6.9)	1.000
Smear positive, *n* (%)	10 (38.5)	45 (44.1)	0.603

^a^ Data are presented as *n* (%). ^b^ Elderly ≥ 65 years old. ^c^ Data are presented as mean ± SD. ^d^ BMI data were available for 126 patients, including 26 with fatal outcomes. OR: odds ratio; CI: confidence interval; Hgb: hemoglobin.

**Table 3 jcm-10-02702-t003:** Risk factors for fatal outcomes.

				Multivariate	
	Fatal ^a^	Favorable ^a^	Univariate	Odds Ratio	
Variables	*n* = 26	*n* = 102	*p*	95% CI	*p*
Female ^a^, *n* (%)	5 (19.2)	29 (28.4)	0.343		
Elderly ^b^, *n* (%)	21 (80.8)	71 (69.6)	0.258		
BMI (kg/m^2^) ^c,d^	20.9 ± 4.4	21.6 ± 4.1	0.467		
BMI (kg/m^2^) < 18.5 ^d^	9 (34.6)	26 (26.0)	0.525		
PG-SGA score ^c^	6.7 ± 4.0	4.8 ± 3.5	0.028	1.142 (1.012–1.288)	0.031
Hgb ^c^	11.5 ± 2.5	11.7 ± 2.6	0.745		
Comorbidity, *n* (%)					
Diabetes mellitus	10 (38.5)	20 (19.6)	0.043	1.993 (0.757–5.248)	0.163
Heart disease	0 (0.0)	2 (2.0)	1.000		
COPD	2 (7.8)	11 (10.8)	1.000		
CKD stage 5	2 (7.8)	4 (3.9)	0.601		
DLC	1 (3.8)	2 (2.0)	0.497		
Solid tumor cancer	8 (30.8)	15 (14.7)	0.083	2.251 (0.785–6.456)	0.131
Hematological malignancies	2 (7.8)	3 (2.9)	0.268		
Constitutional symptom, *n* (%)					
Fever	10 (38.5)	24 (23.5)	0.124		
Body weight loss	2 (7.8)	8 (7.8)	1.000		
Infectious source, *n* (%)					
Pulmonary	25 (96.2)	92 (90.2)	0.460		
Extrapulmonary	2 (7.8)	17 (16.7)	0.360		
Dissemination	1 (3.8)	7 (6.9)	1.000		
Smear positive, *n* (%)	15 (57.7)	40 (39.2)	0.089	1.677 (0.664–4.238)	0.274

^a^ Data are presented as *n* (%). ^b^ Elderly ≥ 65 years old. ^c^ Data are presented as mean ± SD. ^d^ BMI data were available for 126 patients, including 26 with fatal outcomes. OR: odds ratio; CI: confidence interval; Hgb: hemoglobin.

## Data Availability

The data presented in this study are available from the corresponding author upon reasonable request.

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
