# Peer review of "Nutrition Assessment and Adverse Outcomes in Hospitalized Patients with Tuberculosis"

_jcm, 2021, doi:10.3390/jcm10122702_

Round 1

Reviewer 1 Report

The authors present the results after assessing the nutritional status of TB inpatients in a Chinese hospital with a well-validated score (PG-SGA) commonly used in other chronic diseases as cancer. The manuscript is sound and nicely reported. However, I have some suggestions and questions for authors: 

  1. In Methods section 2.1.: I am quite curious on the drug regimen prescribed to patients, that include ethambutol only in some of the patients. Why is that? this is quite strange considering the standard treatment involve the 4 drugs, so I would recommend the authors include an explanation for this. 
  2. In section 2.2.: patients >65 y.o. are considered "elderly" and below this point are considered "non-elderly". However, this "non-elderly" range is too big, in my opinion, and I would recommend to break it down into more groups. Moreover, the authors might see differences in the results of the score if consider more age groups, specially as the concomitant chronic diseases are not as common in young adults than in people on their 50's. 
  3. Only death is considered "unfavourable outcomes", even if according to the WHO Definitions and reporting framework for tuberculosis – 2013 revision (updated December 2014 and January 2020) treatment failed and lost of F-U also are. If the authors do not want to include these, then I would recommend changing "unfavourable outcomes" for death.
  4. Table 1: firs p (female %): a . is missing. What about other symptoms? Are there differences if considering presence of cavities or not? And considering sputum conversion before or after 2 months of treatment regimen? 
  5. Table 3: first row says "hepatitis" and "non-hepatitis". I guess this is hepatotoxicity, right? Also in the text: when authors say "liver injury" mean "hepatotoxicity" by the drug regimen, right?
  6. There are 2 results that are important: 1) the manuscript show that malnourished patients are fatter and have more DM; and 2) cardiac patients and malnourished patients have more hepatotoxicity. This should be better stated, as are the main conclusions. This should also be better discussed taking into account the literature.
  7. Discussion section: I do not see the point in breaking it into subsections. Please restructure it better, as it can be improved.
  8. Include the reference of the WHO guidelines that are mentioned in subsection 4.1.
  9. Please mention if the PG-SGA score have been previously used in TB studies, either in the introduction or in the discussion.
  10. In subsection 4.4., limitations, it is said that the study was retrospective, but it wasn't, as the patients were evaluated when they were admitted in the hospital, right? The data was collected prospectively, and analyzed later, right?
  11. Need to be discussed in the discussion/conclusion section what the authors propose based on their results. Should we use the PG-SGA score for all TB patients? Only in DM/cardiac patients? When? And if used and detecting the malnourished patients, then shall we give nutritional counseling? Please explain and expand.

Author Response

Dear reviewer.

  Thanks for your suggestion and we will submit the new manuscript by your suggestion. We try to reply the suggestion as:

1. In Methods section 2.1.: I am quite curious on the drug regimen prescribed to patients, that include ethambutol only in some of the patients. Why is that? this is quite strange considering the standard treatment involve the 4 drugs, so I would recommend the authors include an explanation for this. 

Reply: If the sensitivity test showed RIP all sensitive, ethambutol may not be included in the therapeutic regiment. We have add the phrase “(if RIP all sensitive)” in the sentence, thanks for your suggestion.

2. In section 2.2.: patients >65 y.o. are considered "elderly" and below this point are considered "non-elderly". However, this "non-elderly" range is too big, in my opinion, and I would recommend to break it down into more groups. Moreover, the authors might see differences in the results of the score if consider more age groups, specially as the concomitant chronic diseases are not as common in young adults than in people on their 50's. 

Reply: Most of our participants were elderly patient (71.9%), and the elderly patients is not included is multivariate analysis for mortality. The case numbers is little in each group if we breaking “non-elderly” participants down into more groups, thanks for your suggestion.

3. Only death is considered "unfavourable outcomes", even if according to the WHO Definitions and reporting framework for tuberculosis – 2013 revision (updated December 2014 and January 2020) treatment failed and lost of F-U also are. If the authors do not want to include these, then I would recommend changing "unfavourable outcomes" for death.

Reply: We have change “unfavourable outcomes” to “fatal outcome”, thanks for your suggestion.

4. Table 1: firs p (female %): a . is missing. What about other symptoms? Are there differences if considering presence of cavities or not? And considering sputum conversion before or after 2 months of treatment regimen? 

Reply: We have current the table 1,thanks for your attention. In this study, we had not recorded presence of cavities and the sputum conversion after 2 months of treatment.

5. Table 3: first row says "hepatitis" and "non-hepatitis". I guess this is hepatotoxicity, right? Also in the text: when authors say "liver injury" mean "hepatotoxicity" by the drug regimen, right?

Reply: In this study “hepatotoxicity” was definition as “liver injury”. We have change to “hepatitis” and “non-hepatitis” to “liver injury” and “non-liver injury”, thanks for your suggestion.

6. There are 2 results that are important: 1) the manuscript show that malnourished patients are fatter and have more DM; and 2) cardiac patients and malnourished patients have more hepatotoxicity. This should be better stated, as are the main conclusions. This should also be better discussed taking into account the literature.

Reply: 1) We discussed with diabetes mellitus and tuberculosis in subsection4.1.

          2) Due to only 2 case with cardiac disease, we emphasized the case number in subsection 3.2.

7. Discussion section: I do not see the point in breaking it into subsections. Please restructure it better, as it can be improved.

Reply: We change the first paragraph in subsection 4.4 to 4.3., thanks for your suggestion

8. Include the reference of the WHO guidelines that are mentioned in subsection 4.1.

Reply: We have added the reference [8] there, thanks for your suggestion.

9. Please mention if the PG-SGA score have been previously used in TB studies, either in the introduction or in the discussion.

Reply: We change the sentence in subsection 4.4. as “Previous study with small case numbers considered the SGA, a nutrition assessment tool, may be a useful for pulmonary TB patients [7].”,and wrote the sentence follow it” This study we firstly evaluated the relationship between PG-SGA score and TB in a single center, and we included all hospitalized patients during the follow-up.”, thanks for your suggestion.

10. In subsection 4.4., limitations, it is said that the study was retrospective, but it wasn't, as the patients were evaluated when they were admitted in the hospital, right? The data was collected prospectively, and analyzed later, right?

Reply: We have re-written the sentence as “First, this study enrolled only inpatients…”, thanks for your suggestion.

11. Need to be discussed in the discussion/conclusion section what the authors propose based on their results. Should we use the PG-SGA score for all TB patients? Only in DM/cardiac patients? When? And if used and detecting the malnourished patients, then shall we give nutritional counseling? Please explain and expand.

Reply: We have add the sentence as “We hope further studies may evaluate the PG-SGA score not only in the initiation tuberculosis therapy but also assess during threaptic course. We the serious assess, we may establish the relationship of therapeutic outcomes including clinical cure, microbiology eradication, adverse events and the dynamic PG-SGA scores of the TB patients.” in the subsection 4.4. thanks for your suggestion.

Reviewer 2 Report

This retrospective study explored the correlation between TB outcomes (favorable or fatal outcome) and nutritional status of patients. BMI, global assessment, and PG-SGA were explored. Although this study is interesting because it provides additional evidence of the importance of taking into account the nutritional status of TB patients in their management, some points have to be clarified.

Abstract, methods sub-section: “Clinical outcomes included clinical responsiveness and mortality.” For “clinical responsiveness”, do the authors mean “liver injury”? If yes, please replace, if no clarify.

Abstract, results sub-section: “A higher PG-SGA score was a risk factor…” Please indicate the threshold of PG-SGA score.

Introduction, first paragraph: Please cite more recent data for the status of TB epidemic.

Introduction second paragraph: Please define the abbreviation PG-SGA correctly.

Methods, subsection 2.2.: Please indicate the thresholds to consider fever and body weight loss.

Methods, subsection 2.4.: Unfavorable outcome in TB usually include treatment failure (delayed culture conversion or culture reversion), relapse or death. Given that the only unfavorable outcome considered in this study was the fatal outcome, please replace “unfavorable outcome” by “fatal outcome” throughout the text.

Results, 3.1.: According to Table 1 the percentage of pulmonary TB was 91.4%. Please modify.

Table 1: There are some mistakes regarding Table 1 footnotes for BMI, PG-SGA score, and hemoglobin. Please modify.

Table 2: Please check that there was no error in the calculation of the odd ratios.

Results, subsection 3.2.: Please indicate the corresponding Tables for these results.

Results, subsection 3.2. first paragraph: BMI was significantly lower, not higher, in malnourishes than in well-nourished patients. Please modify.

Results, subsection 3.2. first paragraph: “PG-SGA scores differed significantly…”. Please clarify.

Results, subsection 3.2. second paragraph: Regarding the ratio of patients with heart disease, please specify that this only concerned two patients, even if the result obtained was significant.

Results, subsection 3.2. second paragraph: How do the authors determined the threshold of PG-SGA ≥6 ? If it was thanks to the ROC curve, please present this result first, otherwise specify.

Figure 1: Please indicate the p-values.

Results, subsection 3.3.: Please indicate the corresponding Tables for these results.

Results, subsection 3.3.: Regarding patients with BMI < 18.5, data do not correspond to those in the Table 2. Please correct.

Results, subsection 3.3.: For the area under the ROC curve for PG-SGA scores compared with outcome, please specify the 95% confidence interval in the main text and in the Figure 2.

Discussion: Please pay attention to references format.

Discussion, first paragraph: Please add references.

Discussion, first paragraph: This study demonstrated a significant, not strong, correlation between PG-SGA scores and TB outcomes.

Discussion, subsection 4.2.: The authors discussed the risk associated with overweight in TB but did not consider patients with a BMI> 25 in their analysis. Please clarify or remove this sentence.

Discussion, subsection 4.4.: The first paragraph would be more appropriate in subsection 4.3. because it does not concern the limitations of the study.

Discussion, subsection 4.4. second paragraph: The authors stated that “the study did not represent all the patients in our hospital with TB” and that they “included all hospitalized patients during the follow-up”. Please clarify.

Discussion, subsection 4.4.: Please discuss other types of unfavorable outcomes useful to stratify TB patients that could be used in future studies to evaluate the usefulness of PG-SGA.

Discussion: Please discuss the added value of PG-SGA compared to global assessment categories, as in both cases there were significant correlations with fatal outcomes and liver injury.

Conclusion: Based on this study, total PG-SGA score and global assessment are significantly, not closely, related to the outcomes of TB and liver injury during anti-TB treatment.

Author Response

Dear reviewer,

  Thanks for your suggestion, we submit the manuscript modified according your suggestion. We reply these suggestion as:

1. Abstract, methods sub-section: “Clinical outcomes included clinical responsiveness and mortality.” For “clinical responsiveness”, do the authors mean “liver injury”? If yes, please replace, if no clarify.

Reply: We have changed the phrase “clinical responsiveness” to “liver injury”, thanks for your suggestion.

2. Abstract, results sub-section: “A higher PG-SGA score was a risk factor…” Please indicate the threshold of PG-SGA score.

Reply: We analyzed the PG-SGA score as continuous variable in table 2 “Risk factors for unfavorable outcomes” and table 3 “Risk factors for liver injury during treatment”. We set the threshold of PG-SGA score as 5.5 according to ROC analysis and wrote the finding in next sentence; thanks for your attention.

3. Introduction, first paragraph: Please cite more recent data for the status of TB epidemic.

Reply: We have re-written the sentence as “Tuberculosis (TB) remains a major worldwide problem. In 2019, an estimated 10.0 million new cases occurred globally and there were 1.4 million TB deaths [1].”and changed the reference, thanks for your suggestion.

4. Introduction second paragraph: Please define the abbreviation PG-SGA correctly.

Reply: We have re-written the phrase as “patient-generated subjective global assessment (PG-SGA)”, thanks for your suggestion.

5. Methods, subsection 2.2.: Please indicate the thresholds to consider fever and body weight loss.

Reply: We have re-written the sentence as “Constitutional symptoms included fever (body temperature ≥ 38゚C) and body weight loss (weight loss ≥ 5% of the initial weight in 6 months).”, thanks for your suggestion.

6. Methods, subsection 2.4.: Unfavorable outcome in TB usually include treatment failure (delayed culture conversion or culture reversion), relapse or death. Given that the only unfavorable outcome considered in this study was the fatal outcome, please replace “unfavorable outcome” by “fatal outcome” throughout the text.

Reply: We have replaced “unfavorable outcome” by “fatal outcome” throughout the text,  thanks for your suggestion.

7. Results, 3.1.: According to Table 1 the percentage of pulmonary TB was 91.4%. Please modify.

Reply: We have corrected the error, thanks for your attention.

8. Table 1: There are some mistakes regarding Table 1 footnotes for BMI, PG-SGA score, and hemoglobin. Please modify.

Reply: We have corrected the error as “ b Data are presented as mean ± SD”, thanks for your suggestion.

9. Table 2: Please check that there was no error in the calculation of the odd ratios.

Reply: We have corrected the error, thanks for your attention.

10. Results, subsection 3.2.: Please indicate the corresponding Tables for these results.

Reply: We have indicated them as “(table 1)” and “(table 3)”, thanks for your suggestion.

11. Results, subsection 3.2. first paragraph: BMI was significantly lower, not higher, in malnourishes than in well-nourished patients. Please modify.

Reply: We have corrected the error, thanks for your attention.

12. Results, subsection 3.2. first paragraph: “PG-SGA scores differed significantly…”. Please clarify.

Reply: We have re-written the sentence as “…and PG-SGA scores also significantly lower in malnourished patients (p < 0.001).”, thanks for your suggestion.

13. Results, subsection 3.2. second paragraph: Regarding the ratio of patients with heart disease, please specify that this only concerned two patients, even if the result obtained was significant.

Reply: We have re-written the sentence as “Comparing the liver-injury-negative group, the liver-injury-positive group had a higher ratio of patients with heart disease (only two participants)(7.7% vs. 0.0%; p = 0.004) and had higher PG-SGA scores (7.3 ± 3.6 vs. 4.7 ± 3.5; p = 0.002) (Table 3)”, thanks for your suggestion.

14. Results, subsection 3.2. second paragraph: How do the authors determined the threshold of PG-SGA ≥6 ? If it was thanks to the ROC curve, please present this result first, otherwise specify.

Reply: We have added the sentence as The ROC analysis for outcome prediction showed that a PG-SGA score of 5.5 points yielded the most appropriate sensitivity (61.5%) and specificity (64.7%).”, thanks for your suggestion

15. Figure 1: Please indicate the p-values.

Reply: We have re-written the sentence as “Moreover, patients with PG-SGA scores ≥ 6 had a significantly higher incidence of liver injury (32.7% vs.11.8%; p = 0.007) and fatal outcomes (30.8% vs. 13.2%; p = 0.024) than did patients with lower PG-SGA scores (Figure 1).”, thanks for your suggestion

16. Results, subsection 3.3.: Please indicate the corresponding Tables for these results.

Reply: We have indicated them as “(table 2)”, thanks for your suggestion.

17. Results, subsection 3.3.: Regarding patients with BMI < 18.5, data do not correspond to those in the Table 2. Please correct.

Reply: We have corrected the error, thanks for your attention.

18. Results, subsection 3.3.: For the area under the ROC curve for PG-SGA scores compared with outcome, please specify the 95% confidence interval in the main text and in the Figure 2.

Reply: We have added 95% confidence interval: 0.530-0.767, thanks for your attention.

19. Discussion: Please pay attention to references format.

Reply: We have modified the references format, thanks for your attention.

20. Discussion, first paragraph: Please add references.

Reply: We have re-written the paragraph as “…Malnutrition as smoking, alcohol abuse, diabetes and Human Immunodeficiency Virus infection has contribution to the attributable risk for tuberculosis; and the consequences of malnutrition may include decreased response to treatment, a delay in recovery, a compromised immune system, increased susceptibility to infection, a lower quality of life, and an increased risk of death in many patients [15]...” and add the reference, thanks for your suggestion.

21. Discussion, first paragraph: This study demonstrated a significant, not strong, correlation between PG-SGA scores and TB outcomes.

Reply: We have re-written the phrase, thanks for your suggestion.

22. Discussion, subsection 4.2.: The authors discussed the risk associated with overweight in TB but did not consider patients with a BMI> 25 in their analysis. Please clarify or remove this sentence.

Reply: We have removed the sentence, thanks for your suggestion.

23.Discussion, subsection 4.4.: The first paragraph would be more appropriate in subsection 4.3. because it does not concern the limitations of the study.

Reply: We have moved the paragraph to 4.3, thanks for your suggestion.

24. Discussion, subsection 4.4. second paragraph: The authors stated that “the study did not represent all the patients in our hospital with TB” and that they “included all hospitalized patients during the follow-up”. Please clarify.

Replay: We have re-written the sentence as”… ,and the participants diagnosis in outpatient department were not be included.” , thanks for your suggestion.

25. Discussion, subsection 4.4.: Please discuss other types of unfavorable outcomes useful to stratify TB patients that could be used in future studies to evaluate the usefulness of PG-SGA.

Replay: We have re-written the sentence as “We hope further studies may evaluate the PG-SGA score not only in the initiation tuberculosis therapy but also assess during threaptic course. We the serious assess, we may establish the relationship of therapeutic outcomes including clinical cure, microbiology eradication, adverse events and the dynamic PG-SGA scores of the TB patients.”

26. Discussion: Please discuss the added value of PG-SGA compared to global assessment categories, as in both cases there were significant correlations with fatal outcomes and liver injury.

Reply: We have added the sentence as “Comparing global assessment categories, PG-SGA is a continuous measure with multiple-level analysis for nutrition status.” in subsection 4.4, thanks for your suggestion.

27. Conclusion: Based on this study, total PG-SGA score and global assessment are significantly, not closely, related to the outcomes of TB and liver injury during anti-TB treatment.

Reply: We have re-written the phrase, thanks for your suggestion.